# Deep Temporal Deaggregation: Large-Scale Spatio-Temporal Generative Models

## Abstract

Access to spatio-temporal trajectory data is essential for improving infrastructure, preventing the spread of disease and for building autonomous vehicles. However, it remains underutilized due to limited availability, as it cannot be shared publicly due privacy concerns or other sensitive attributes. Generative time-series models have shown promise in generating non-sensitive data, but show poor performance for large-scale and complex environments. In this paper we propose a spatio-temporal generative model for trajectories, TDDPM, which outperforms and scales substantially better than state-of-the-art. The focus is primarily on trajectories of peoples' movement in cities. We propose a conditional distribution approach which unlock out-of-distribution generalization, such as to city-areas not trained on, from a spatial aggregate prior. We also show that data can be generated in a privacy-preserving manner using $k$-anonymity. Further, we propose a new comprehensive benchmark across several standard datasets, and evaluation measures, considering key distribution properties.

## 1 Introduction

Time-series data of human mobility enables applications such as pandemic forecasting and management (Ilin et al., 2021), smart city development (Wang et al., 2022), urban governance (Xiong et al., 2024), human rights violation detection (Tai et al., 2022) and monitoring of global migration induced by war and climate change (Niva et al., 2023; Alessandrini et al., 2020). Two major challenges stand in the way for using time-series data to these ends. The first is a *shortage of publicly available data* (Ansari et al., 2024). Data can only be collected and shared in limited capacity due to privacy concerns, business concerns and national security, creating a silo effect. Secondly, *predictions* about unobserved parts in space, about the future or even possible futures - given that different actions are taken - is often a necessary complement to the readily collectable data. One such case is generating high-fidelity realistic spatio-temporal trajectory data, such as individual pedestrians navigating a city or a building. Open problems within the road traffic domain (Lana et al., 2018), and using human mobility data at large, is (1) high quality large-scale trajectories, (2) adaptation to sudden environmental changes and (3) facilitation of what-if analysis for environmental changes.

A solution to both challenges is to use time-series generative models to accurately capture the data distribution. These generative models can then be adapted to generate samples that are private (Yoon et al., 2019b; Wang et al., 2023), resulting in synthetic non-private datasets that can be made publicly available. Further adaptation allows for tasks such as imputation or forecasting (Alcaraz & Strodthoff, 2023). However, current approaches have crucial limitations limiting their real-world applicability: they can only generating very short sequence length and they struggle to model complex distributions with sparse support, including even the smaller of spatio-temporal datasets used in this work.

In this work, *we introduce a new method for generating high-fidelity trajectory data for complex large-scale environments, Temporal Denoising Diffusion Probabilistic Model (TDDPM),*

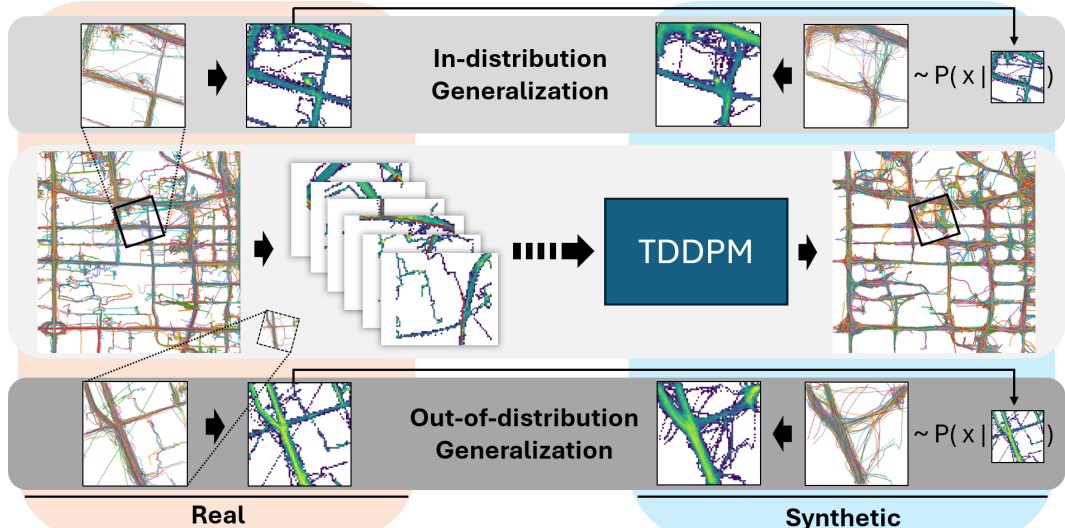

Figure 1: TDDPM is trained on real 2D trajectories (left) to generate synthetic trajectories (right), conditioned on how likely it should be for the population of synthetic trajectories to occupy space on the 2D plane. The latter is represented as a discrete distribution over occupancy frequency, i.e., the marginal distribution over the trajectory probability distribution if integrating out time. This yields both high-fidelity in-distribution generalization (top) and out-of-distribution generalization (bottom), the latter when conditioning on a marginal distribution not part of the training data (dashed rectangle).

capable of out-of-distribution generalization via deaggregation from spatial statistics to temporal samples, i.e. trajectories. More specifically, we first adapt the denoising diffusion model to generate time-series data. Secondly, we ask: (1) *Can a deep generative model (in our case a diffusion model) be made to stay true to a marginal probability distribution, while generating samples from the full distribution?* and (2) *How can out-of-distribution generalization be achieved, to support synthetic data generation of new environments or hypothetical scenarios?* We demonstrate how this can be achieved via conditioning the model on local information. More specifically, on a spatial aggregate prior which is a non-temporal marginal distribution over a set of trajectories, making deaggregation to individual trajectories possible. Figure 1 illustrates tasks for and capabilities of TDDPM in a setting of mobility trajectory data in a big city.

Our main contributions are:

- We propose TDDPM, a trajectory generation method which can generate high-quality synthetic trajectory data while accurately capturing intricate spatial and temporal distributions, scaling effectively across datasets of varying size and complexity. TDDPM implements a novel deaggregation technique, leveraging spatial aggregate priors, which also enables generation of synthetic data for new, unseen environments.

- We harmonize different notions of synthetic data quality and design a comprehensive benchmark to thoroughly evaluate performance on the unconditional synthetic trajectory data task, for large-scale and complex environments.

- We demonstrate that TDDPM achieve state-of-the-art on this task, with unprecedented levels of quality both on sample level, and distribution-wise. We also demonstrate that the novel deaggregation architecture of TDDPM achieve out-of-distribution generalization performance similar to in-distribution generalization.

- We show that TDDPM's novel design makes it straight-forward to apply TDDPM to tasks such as synthetic data generation for new, unseen environments, and hypothetical scenarios, as well as to privacy-preserving generation using k-anonymity.

## 2 PROBLEM DEFINITION

The task of learning an unconditional generative model is defined as learning a mapping $f$ from samples drawn from a known distribution $\mathcal{D}_{\text{known}}$, e.g. standard normal distribution, to samples from an unknown target distribution $\mathcal{D}_{\text{unknown}}$. The mapping is learned without direct access to the unknown distribution, and is instead limited to a set of samples $X_{\text{train}} = \{x_1, \ldots, x_N\}$, where $x_i \sim \mathcal{D}_{\text{unknown}}$. Once the mapping has been learned, synthetic data can be generating by first sampling the known distribution, then passing the individual samples through the mapping function $X_{\text{synthetic}} = \{f(y_i)\}_{i=0}^M$, where $y_i \sim \mathcal{D}_{\text{known}}$. The goal of this mapping is for the synthetic samples to be *similar* to samples from the known distribution.

## 3 RELATED WORK

Previous work on unconditional generation of time-series data has focused on variations of the generative adversarial networks (GAN) architecture (Esteban et al., 2017; Yoon et al., 2019a; Jeon et al., 2022) and, more recently diffusion models (Zhu et al., 2023; Yuan & Qiao, 2024). See Appendix A.1 for a short description of the GAN-based architectures. There has also been an interest in using time-series generation for imputation and forecasting (Alcaraz & Strodthoff, 2023; Tashiro et al., 2021; Shen & Kwok, 2023; Dai et al., 2023; Feng et al., 2024). Transformer-based time-series foundation models has been proposed as a general purpose forecasting tool (Ansari et al., 2024), but has not been evaluated on the unconditional generation task.

DiffTraj (Zhu et al., 2023) adapts the denoising diffusion architecture (Ho et al., 2020) to generate GPS-trajectories. The U-Net architecture used in (Ho et al., 2020) for denoising, is also adapted to work with trajectories. As part of the generation process, the model is conditioned on individual trajectory-specific statistics such as velocity, distance, and departure time.

Diffusion-TS (Yuan & Qiao, 2024) also adapts the denoising diffusion architecture (Ho et al., 2020) to generate time-series data by implementing the denoising step with a multilayer neural network each consisting of a transformer block, a fully connected neural network as well as time-series specific layers with the aim of improved performance and interpretability. The architecture also has support for conditional generation to enable using the models for imputation and forecasting.

As Diffusion-TS move away from U-Net-approach in DiffTraj to a partial Transformer architecture, we extend this further to a full Transformer-based architecture. By doing so, we can simultaneously avoid the additional time-series specific induction bias in Diffusion-TS while gaining increased performance and capabilities for synthetic data generation. While both DiffTraj and Diffusion-TS have support for using conditional information in their models, our approach differs both in terms of flexibility and in the purpose. The conditional modeling in Diffusion-TS is not for increased sample quality, but instead for enabling additional capabilities for additional tasks, namely forecasting and imputation, which is different than synthetic data generation, and cannot be directly adapted for use of local information.

In DiffTraj, conditional information is used to improve sample quality. Our approach also use conditional information to improve performance, the difference is the type of information used for the conditioning. We use statistics related to the local environment, which we refer to as *local information*, which in general cannot be directly tied to any individual sample. In DiffTraj the statistics are directly related to individual samples, which in practice create a one-to-one mapping between sample and conditional. The use of local information allows our method to generalize to entirely new environments and scenarios, while conditioning on sample-level statistics only allows for generating samples within the same spatial region trained on. Conditioning on information that is sample-specific also increase the risk of memorization, as the model might simply learn to store and retrieve individual training data points (trajectories) when conditioning on the individual statistic of any single training data point. In our work we use distinctly different conditional information for training and sampling for evaluation, making our method less susceptible to memorization issues.

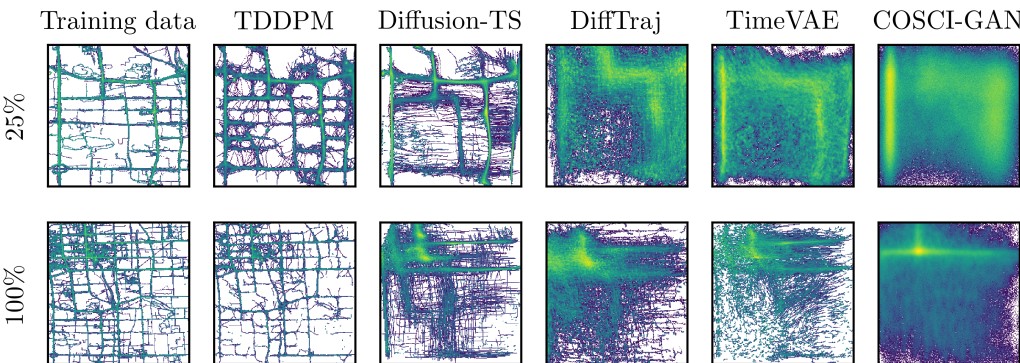

Figure 2: Comparision of the heatmap of the training data and the resulting synthetic data for the models compared in this paper. The training data is taken from Geolife and covers 40 and $161\,km^2$ respectively. TDDPM, our proposed method, scales successfully from the smaller region to the larger and without re-training. Meanwhile, the other models have to be re-trained for each dataset and are mostly unable to capture the training data faithfully.

Outside the field of unconditional time-series generation, the transformer architecture (Vaswani et al., 2017) demonstrated that attention as viable and flexible alternative to recurrence in neural networks and the architecture itself became a common architecture. The vision transformer (ViT) (Dosovitskiy et al., 2020) is based on the transformer encoder and splits images into several equally sized patches which are projected before given as input tokens to the encoder.

In practice, the non-temporal marginal distributions are realized as heatmaps. They are processed similar to how ViT processes images (Dosovitskiy et al., 2020), by splitting the heatmaps into 8x8 equally sized patches and linearly projecting the patches into tokens.

## 4 TEMPORAL DENOISING DIFFUSION PROBABILISTIC MODEL (TDDPM)

We propose the decomposition of unconditional generation into two separate steps: (1) calculating an information rich, interpretable and concise aggregation of the available data and (2) use a conditional generative model to de-aggregate into synthetic samples. More precisely, instead of learning to generate samples from $p(x)$ directly, we learn to generate samples from the joint distribution $p(x, l)$, where $l$ is aggregates of local information. This is achieved via the chain rule: $p(x, l) = p(x|l)\, p(l)$, where $p(x|l)$ is a conditional deep generative model and $p(l)$ can be modeled using a explicit and interpretable distribution. This allows for more accurate and representative modelling on challenging distributions, as well as enabling out-of-distribution generalization when $l$ include sufficient information.

Sampling $p(x)$ is done by first sampling $p(l)$ and then using these samples to sample the conditional distribution $p(x|l)$. In practice, we learn a conditional mapping function $g : \mathcal{D}_{\text{known}} \times \mathcal{L} \rightarrow \mathcal{D}_{\text{unknown}}$, which maps samples from a known distribution $D_{\text{known}}$ and local information $l \in \mathcal{L}$ to synthetic samples, which should be *similar* to samples from the unknown distribution $\mathcal{D}_{\text{unknown}}$. For a collection of local information $L = \{l_i\}_{i=0}^{M}$, a synthetic dataset is generated by $X_{\text{synthetic}} = \left\{ g(y_i, l_i) \right\}_{i=0}^{M}$, where $y_i \sim \mathcal{D}_{\text{known}}$. For certain problems, if the choice of $l$ is sufficiently informative, we can learn to generate data for a new distribution $p^*(x)$ without having to retrain $p(x|l)$. This is achieved by estimating a new distribution $p^*(x, l)$, s.t. $p^*(x, l) \approx p(x|l)\, p^*(l)$. This is not only a computationally efficient way to estimate $p^*(x)$, but can also be useful in cases where we only have access to $l$, e.g. to model scenarios or account for data distribution shifts.

### 4.1 OCCUPANCY FREQUENCY MARGINAL DISTRIBUTION AS LOCAL INFORMATION

We define local information $\ell$ as follows. Let $\mathcal{G}$ be a grid where each cell has width and height $\Delta$. Each cell in $\mathcal{G}$ is indexable by $i, j$ for $i = 0, \ldots, N$ and $j = 0, \ldots, J$ such that $\mathcal{G}[i, j]$ is the cell on the $i$th row and $j$th column on the grid. The grid $\mathcal{G}$ has a position, rotation and area in the world, such that grid cell $\mathcal{G}[i, j]$ has position $\mathrm{p}_{i,j}$, rotation $\theta_{i,j}$ and area $\Delta^2$ m$^2$.

Given sequences $x[n] \in \mathbb{X}$, $n = 1, \ldots, N$. We want to calculate a relative occupancy on the grid, defined as the ratio of $x[n] \in \mathbb{X}$ that is inside each grid cell as opposed to outside of the grid cell. The probability for an observation $x$ to be present inside $\mathcal{G}[i, j]$ is therefore

$$\ell(\mathrm{p}, \theta, \Delta) \stackrel{\text{def}}{=} p(x \in \mathcal{G}[i, j]) \propto \sum_{x[n] \in \mathbb{X}} \sum_{n} \mathbb{1}\Big(x[n] \in \mathcal{G}[i, j]\Big), \tag{1}$$

where $\mathbb{1}(\cdot)$ is the indicator function which is 1 if the condition is true and 0 otherwise. This constitutes a discrete probability distribution on the occupancy frequency over the grid cells, as a marginal distribution over time, $n$, and an aggregation over space, $\mathcal{G}$ with resolution $\Delta$.

### 4.2 HIERARCHICAL OCCUPANCY FREQUENCY MIXTURE

To increase fidelity at large scale and to model macro-scale proportionality in spatial occupancy frequency, we learn a mixture model over the cells of rigid grid over the training data support. Let $\mathcal{H}$ be a grid where each grid-cell $H[a, b]$ consists of an individual sub-grid $\mathcal{G}_{a,b}$, each of which has $\theta = \mathbf{0}$ and $64 \times 64$ grid cells of size $\Delta$.

$$p(x \in \mathcal{H}[a, b]) \propto \sum_{x[n] \in \mathbb{X}} \sum_{n} \mathbb{1}\Big(x[n] \in \mathcal{H}[a, b]\Big), \tag{2}$$

The mixture model is defined as

$$p(x) = \sum_{a} \sum_{b} p(x \in \mathcal{H}[a, b]) \ell(\mathrm{p}_{a,b}, \mathbf{0}, \Delta), \tag{3}$$

where $p(x \in \mathcal{H}[a, b])$ is the mixture weight for component $(a, b)$ and $\ell(\mathrm{p}_{a,b}, \mathbf{0}, \Delta)$ is the occupancy frequency distribution for region $(a, b)$.

### 4.3 ARCHITECTURE

To approximate $p(x|l)$ we propose an architecture based on the denoising diffusion architecture (Ho et al., 2020), using a transformer encoder (Vaswani et al., 2017) for denoising. At each denoising step, the transformer takes the entire noisy sequence, the current denoising step and the local information as input and predicts the noise added at the current step. An overview of the approach is shown in Figure 3.

We set $l$ to be a local occupancy map of the training data trajectories for a given region $r = (x, y, \theta)$ where $\theta$ is rotation in $[0, 2\pi)$. The occupancy map contain information about relative frequency, i.e. where which areas in the regions are more frequently visited and which areas are never visited. They are calculated using all available training data in the given region and we set the dimensions to be $64 \times 64$. The occupancy maps are tokenized similar to the vision transformer (Dosovitskiy et al., 2020), by splitting the occupancy map into $8 \times 8$ sub-maps and then linearly projecting them. See Appendix A.2 for full details.

### 4.4 TRAINING

We extract a training data set by uniformly sampling regions (including their rotation) and, for each region, saving a tuple consisting of an occupancy map of the region and a scaled sub-trajectory from the region. The sub-trajectories are transformed subtracting the regions origin point and by rotating it accordingly. If the sub-trajectory is longer than the target sequence length, a random sub-sequence is chosen. All regions have the same size, which corresponds to approximately 1% of the total area used for training. See Figure 1 for examples of occupancy maps and trajectories.

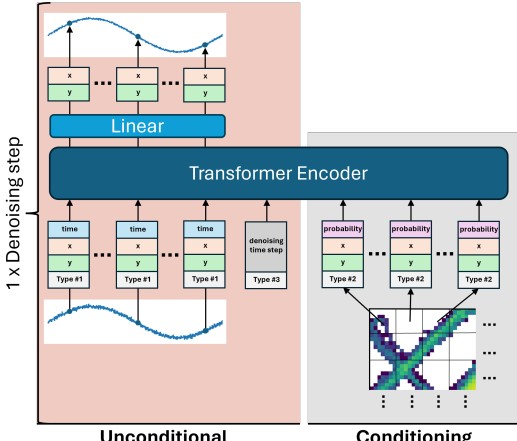

Figure 3: Overview of the architecture. In the unconditional part, each time point of the noisy trajectory is converted into a separate token with positional embedding (Vaswani et al., 2017) used to embed its values and the time point, as well as a learned vector representing its type. The denoising step token encodes the denoising step, the step is encoding using positional encoding and then concatenated with a type vector. In addition to trajectories, the transformer also take a marginal distribution to guide the denoising process to generate samples with particular properties, improving in-distribution performance as well as enabling generalization to previously unobserved areas. The marginal distribution is split into tokens, taking inspiration from ViT (Dosovitskiy et al., 2020), concatenated with a learned type vector and the corresponding position using positional embedding.

## 4.5 Sampling

The target region is split up into several subregions. For each subregion, an occupancy map is calculated and the number of total users in that subregion is counted. The occupancy map is calculated by dividing the region into a set of cells, counting the number of observations in each cell and lastly normalizing it. The model is then sampled once for each subregion, passing the occupancy map and drawing a number of trajectories linearly proportional to the total number of observations in the region.

When sampling the model, we follow the original sampling schema for denoising diffusion probabilistic models (Ho et al., 2020). Note the full occupancy map is available to the model throughout the sampling process to help guide it and that no noise is added to it. The occupancy map can also be manually modified to model hypothetical scenarios, such as a road being removed or even moved.

## 5 Evaluation

In this section, we start with a comparative study on the unconditional generation task. We compare with state-of-the-art methods from all three major paradigms: GAN, VAE and Diffusion. These include TimeGAN (Yoon et al., 2019a), TimeVAE (Desai et al., 2021), COSCI-GAN (Seyfi et al., 2022), DiffTraj (Zhu et al., 2023) and DiffusionTS (Yuan & Qiao, 2024). We compare across two large-scale GPS-based human trajectory datasets and across several evaluation measures, which together capture all wanted properties.

We then show how our proposed method can generalize to new unseen environments. First by training on a limited part of the map and, using only local information, generating for the remaining map. Secondly by synthetic local information, showcasing the capability of our approach to answer what-if questions about hypothetical environments. Current approaches in the literature are incapable of either. Lastly, we perform an ablation study on the use of local information.

**Evaluation Measures**

There are several perspectives on synthetic data quality in the literature (Alaa et al., 2022; Wu et al., 2021; Esteban et al., 2017) covering a range of partially-overlapping aspects. In this work, we propose to harmonize these and consequently span high fidelity, covering the support, moving beyond copying the training data, downstream task informativeness and distributional proportionality, as captured by the following five qualities:

(I) *Fidelity:* (Alaa et al., 2022) The individual synthetic samples should have similar characteristics to, or be indistinguishable from, samples from the original distribution.

(II) *Diversity:* (Alaa et al., 2022) It should be possible for synthetic data to be drawn from any part of the unknown distribution's support.

(III) *Proportionality:* (Wu et al., 2021) The probability of a sample occurring in the synthetic distribution should be proportional to the probability of a sample occurring in the unknown distribution.

(IV) *Usefulness:* (Esteban et al., 2017) The synthetic data should capture aspects of the unknown distribution that is useful for downstream tasks.

(V) *Generalization:* (Alaa et al., 2022) Synthetic samples should not be copies of the training data.

As there is not a single measure that can encapsulate all qualities, we propose the use of the following evaluation measures to paint a fuller picture of the synthetic data quality:

- **TSTR:** Train on synthetic, test on real. We follow the evaluation done by TimeGAN (Yoon et al., 2019a) and use the synthetic data to train a GRU-based RNN on task of one step prediction using the synthetic data. The resulting model is then evaluated on the training data and the mean absolute value is reported. This evaluates the usefulness, fidelity and diversity of the synthetic data. Lower is better.

- **KL Divergence:** Evaluates support coverage as well as proportionality. $\mathrm{KL}(\mathrm{real} \| \mathrm{synthetic})$ measures how well the synthetic distribution represents the real distribution, with 0 being identical. $\mathrm{KL}(\mathrm{synthetic} \| \mathrm{real})$ measures how well the synthetic distribution fits inside the real distribution, i.e. it is sufficient that the synthetic distribution matches a single mode of the real distribution in order to get a low divergence value. In the latter case, the synthetic distribution consequently do not need to have the same support, nor have any probability density outside of this mode. Symmetric KL weight these two together and Jensen–Shannon divergence is a more stable version of symmetric KL with regions without probability in either of the distributions.

We also make use of the sample-level measures **Density Error**, **Trip Error**, **Length Error** and **Pattern Score**, introduced in Zhu et al. (2023) (see A.3.3 for a high-level description).

We use two publicly real-world GPS-trajectory datasets to run the evaluation: Geolife (Zheng et al., 2011) and Porto (Moreira-Matias et al., 2013). Geolife consists of data from 182 users collected over three years, resulting in 17,621 trajectories mostly centered around Beijing, China. Porto was originally released as part of the Taxi Service Trajectory challenge and contains data on 442 taxis in Porto, Portogal. The data was collected over more than a year. For both datasets, we use only time, longitude and latitude. The data is also filtered for the time to be approximately even between samples, one observation every 10 seconds for Geolife and one every 15 second for Porto.

## 5.1 LARGE-SCALE UNCONDITIONAL TRAJECTORY GENERATION

In this experiment, we evaluate our proposed method on the unconditional generation problem for spatio-temporal data. For each dataset we use two different subsets, a smaller region and a larger region, to scale the difficulty of the problem. The smaller regions are denoted 25% and correspond to the the lower left quarter of the larger subsets. For each dataset and size, we train each model from scratch and then sample them to generate a synthetic dataset used for evaluation. The synthetic datasets are then evaluated with a

range of evaluation measures and the resulting measures are presented in Table 1. The spatial occupancy maps of the synthetic data and the training data are presented in Figure 2 (except for the second row of TDDPM). For a more detailed view of the synthetic data, see Appendix A.3.5. DiffTraj is used in unconditional mode since we (1) disallow the use of identical information between training and evaluation, (2) disallow individual sample-specific conditional information and (3) the source code for conditional mode is missing.

The proposed method is less performant on TSTR on Geolife but the best performer on Porto. This is in contrast to ocular inspection (Appendix A.3.5) of the generated trajectories, where the proposed method and DiffusionTS produce synthetic data of clearly higher quality than the other approaches. On all the distributional distributional similarity measures, the proposed method is well ahead of the related work. TDDPM excels at support coverage and proportionality.

## 5.2 GENERALIZING TO NEW ENVIRONMENTS AND HYPOTHETICAL SCENARIOS

To evaluate the generalization capability of our proposed method, we generate heatmaps outside of the area used for training the model. In regions outside the geographical area, the heatmaps are generated with data previously not observed by the model. The model is sampled once for each heatmap, the number of samples drawn proportional to the total number of observations in the area of the heatmap. Figure 2 for TDDPM, second row, show the synthetic data for out-of-distribution generalization from the bottom-left quarter of the training data. To evaluate the quality of the resulting synthetic data, we run it through the same evaluation suite as previously. The resulting measures are shown in Table 2 and baselines are presented in Table 1. We observe that TDDPM beats state-of-the-art even while generalizing from a fraction of the environment, where the previous state-the-art have access to data from the full environment. We also provide a detailed view of the samples and the resulting heatmaps in Appendix A.3.5. Finally, a proof of concept of what-if-scenario modeling is shown in Appendix A.3.1.

The results in Table 2 indicate that the out-of-distribution generalization with the local information acts as a regularizer, with better performance overall. $\text{KL}(S \parallel R)$ being higher for 100% than for 25% indicate a higher mode-fit for the 100% model while $\text{KL}(S \parallel S)$ being lower for 100% indicate a higher degree of generalization in terms of support and proportionality of the 25% model.

## 5.3 PRIVATE SYNTHETIC TRAJECTORIES WITH k-ANONYMITY

In this experiment, we apply $k$-anonymity (Samarati & Sweeney, 1998) to the local information to generate privacy-preserving synthetic trajectory data. We use the model trained on 25% of the data and generate data for the full 100%. We protect the privacy of the users in the previously unseen environment by applying $k$-anonymity when extracting the local information. More precisely, for each cell in the spatial occupancy map we set cells to zero if we count fewer than $k$ users in corresponding area. We run the experiment for a range of different values for $k$, see Table 3 and for baselines see Table 1. We see that the performance remains relatively consistent for increasing $k$-values. The spatial occupancy maps of the synthetic data are shown in Figure 4. We observe that the quality of the synthetic data is maintained even with higher degrees of privacy, causing only small increases the measures. The quality decrease mostly affects areas with fewer users, as we might only have a few observations for a given street. However, this does not seem to affect downstream performance. In summary, the proposed approach maintain high synthetic data quality even under strict privacy constraints. See A.3.2 for a more detailed elaboration on the privacy protection of our model.

### 5.3.1 ABLATION STUDY

In this experiment, we analyze the effect of conditioning on local information over having access to no local information. TDDPM operate in a two-step process, where the first step is to construct local information and the second step draw trajectories (samples) by conditioning on the local information. TDDPM without local information does no internal conditioning

Table 1: Evaluation of different models' performance across several datasets and measures.

| Measure | Model | Geolife 25% | Geolife 100% | Porto 25% | Porto 100% |
|---|---|---|---|---|---|
| TSTR (↓) | TimeGAN | $0.151 \pm .089$ | $0.107 \pm .096$ | $0.064 \pm .081$ | $\mathbf{0.065 \pm .049}$ |
| | TimeVAE | $\mathbf{0.120 \pm .070}$ | $\mathbf{0.089 \pm .084}$ | $0.063 \pm .076$ | $0.066 \pm .048$ |
| | COSCI-GAN | $0.121 \pm .069$ | $0.090 \pm .090$ | $0.072 \pm .081$ | $0.076 \pm .057$ |
| | Diffusion-TS | $\mathbf{0.120 \pm .070}$ | $0.102 \pm .097$ | $0.066 \pm .081$ | $0.066 \pm .052$ |
| | DiffTraj | $0.137 \pm .100$ | $0.092 \pm .078$ | $0.071 \pm .076$ | $0.118 \pm .067$ |
| | **TDDPM** | $0.148 \pm .099$ | $0.127 \pm .089$ | $\mathbf{0.061 \pm .073}$ | $\mathbf{0.065 \pm .049}$ |
| KL($S \parallel R$) (↓) | TimeGAN | 3.617 | 2.712 | 2.755 | 2.955 |
| | TimeVAE | 1.891 | 2.419 | 2.324 | 2.316 |
| | COSCI-GAN | 2.079 | 2.546 | 2.398 | 2.466 |
| | Diffusion-TS | 1.049 | 2.048 | 2.505 | 2.633 |
| | DiffTraj | 2.310 | 2.909 | 2.483 | 3.521 |
| | **TDDPM** | **0.671** | **0.419** | **2.040** | **0.851** |
| KL($R \parallel S$) (↓) | TimeGAN | 3.000 | 2.175 | 2.645 | 2.497 |
| | TimeVAE | 1.531 | 1.821 | 2.084 | 1.739 |
| | COSCI-GAN | 1.802 | 1.922 | 2.225 | 1.860 |
| | Diffusion-TS | 1.155 | 1.542 | 2.111 | 2.076 |
| | DiffTraj | 2.069 | 2.287 | 2.461 | 2.906 |
| | **TDDPM** | **0.524** | **0.434** | **1.797** | **1.338** |
| KL$_{\mathrm{sym}}$ (↓) | TimeGAN | 3.309 | 2.443 | 2.700 | 2.726 |
| | TimeVAE | 1.711 | 2.120 | 2.204 | 2.028 |
| | COSCI-GAN | 1.940 | 2.234 | 2.311 | 2.163 |
| | Diffusion-TS | 1.102 | 1.795 | 2.308 | 2.354 |
| | DiffTraj | 2.189 | 2.598 | 2.472 | 3.213 |
| | **TDDPM** | **0.597** | **0.426** | **1.919** | **1.095** |
| JS (↓) | TimeGAN | 0.472 | 0.378 | 0.411 | 0.399 |
| | TimeVAE | 0.318 | 0.337 | 0.359 | 0.326 |
| | COSCI-GAN | 0.353 | 0.370 | 0.375 | 0.345 |
| | Diffusion-TS | 0.195 | 0.298 | 0.352 | 0.358 |
| | DiffTraj | 0.384 | 0.393 | 0.391 | 0.463 |
| | **TDDPM** | **0.119** | **0.088** | **0.299** | **0.193** |
| Density (↓) | TimeGAN | 0.320 | 0.201 | 0.163 | 0.132 |
| | TimeVAE | 0.141 | 0.060 | 0.073 | 0.051 |
| | COSCI-GAN | 0.197 | 0.139 | 0.116 | 0.102 |
| | Diffusion-TS | 0.058 | 0.061 | **0.061** | 0.056 |
| | DiffTraj | 0.229 | 0.175 | 0.172 | 0.406 |
| | **TDDPM** | **0.037** | **0.029** | 0.197 | **0.036** |
| Trip (↓) | TimeGAN | 0.362 | 0.238 | 0.166 | 0.154 |
| | TimeVAE | 0.171 | 0.081 | 0.117 | 0.080 |
| | COSCI-GAN | 0.249 | 0.166 | 0.193 | 0.166 |
| | Diffusion-TS | 0.086 | 0.076 | **0.076** | 0.068 |
| | DiffTraj | 0.246 | 0.185 | 0.169 | 0.363 |
| | **TDDPM** | **0.070** | **0.042** | 0.203 | **0.047** |
| Length (↓) | TimeGAN | 0.027 | **0.013** | 0.075 | 0.017 |
| | TimeVAE | 0.459 | 0.072 | 0.349 | 0.201 |
| | COSCI-GAN | 0.579 | 0.729 | 0.677 | 0.612 |
| | Diffusion-TS | **0.014** | 0.020 | **0.014** | **0.010** |
| | DiffTraj | 0.231 | 0.301 | 0.459 | 0.437 |
| | **TDDPM** | 0.075 | 0.106 | 0.105 | 0.105 |
| Pattern (↑) | TimeGAN | 0.500 | 0.700 | 0.710 | 0.760 |
| | TimeVAE | 0.630 | 0.760 | 0.740 | **0.840** |
| | COSCI-GAN | 0.540 | 0.680 | 0.660 | 0.780 |
| | Diffusion-TS | 0.890 | 0.820 | 0.760 | 0.830 |
| | DiffTraj | 0.570 | 0.730 | 0.620 | 0.420 |
| | **TDDPM** | **0.900** | **0.860** | **0.780** | **0.840** |

Table 2: Generalization experiment.

| Target | Trained on | TSTR | KL($S \parallel R$) | KL($R \parallel S$) | KL$_{sym}$ | JS |
|---|---|---|---|---|---|---|
| Geolife | 100% | $0.127 \pm 0.089$ | **0.419** | 0.434 | 0.426 | 0.088 |
| | 25% | $\mathbf{0.094 \pm 0.076}$ | 0.453 | **0.378** | **0.416** | **0.085** |

(the gray part at the right of Figure 3 is ignored). The results are shown in Table 4, which demonstrate a clear improvement using the two-step process (with conditioning).

Table 3: Evaluation of privacy-preserving trajectory data generation. $k$-anonymity has been achieved by filtering out areas with fewer than $k$ users by setting the local information to 0. Lower score corresponds to better performance.

| $k$-privacy | TSTR | KL$(S \parallel R)$ | KL$(R \parallel S)$ | KL$_{sym}$ | JS |
|---|---|---|---|---|---|
| $k = 0$ | **$0.088 \pm 0.086$** | 0.453 | **0.378** | **0.416** | **0.085** |
| $k = 5$ | $0.089 \pm 0.086$ | 0.453 | 0.389 | 0.421 | 0.086 |
| $k = 10$ | $0.089 \pm 0.087$ | 0.448 | 0.391 | 0.420 | 0.086 |
| $k = 20$ | **$0.088 \pm 0.086$** | **0.444** | 0.403 | 0.424 | 0.087 |

| Training data | $k = 0$ | $k = 5$ | $k = 10$ | $k = 20$ |
|---|---|---|---|---|

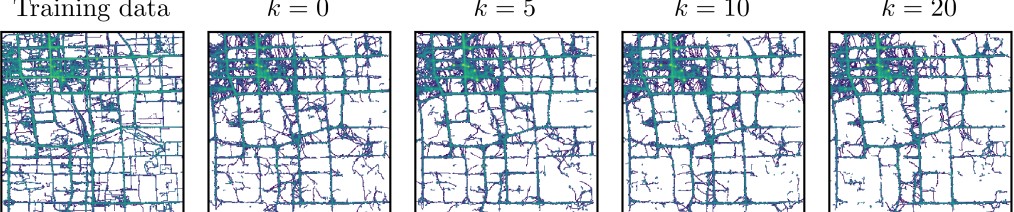

Figure 4: Spatial occupancy maps of the training data, as well as synthetic data generated with increasing levels of privacy. Privacy is achieved using $k$-anonymity by removing local information corresponding to fewer than $k$ users. Increasing the privacy decreases the quality around the less travelled roads, but affects more travelled roads less.

## 6 CONCLUSION

Time-series generative models for trajectory data is a challenging problem. By decomposing the unconditional task into aggregating local information and then deaggregate into high-fidelity trajectories, we demonstrate that TDDPM scales to problem sizes far beyond current state-of-the-art, without compromising fidelity in the generated trajectories. By applying $k$-anonymity on the local information, we also show that high-quality trajectories can be shared without compromising on privacy. Finally, high-quality out-of-distribution generalization is demonstrated at scale, including for what-if scenario modeling when a road is moved.

Table 4: Ablation study of the effect of local information.

| Measure | Model | Geolife 25% | Geolife 100% | Porto 25% | Porto 100% |
|---|---|---|---|---|---|
| TSTR ($\downarrow$) | TDDPM | $0.148 \pm 0.099$ | **$0.127 \pm .089$** | **$0.061 \pm .073$** | **$0.065 \pm .049$** |
| | w/o local info | **$0.119 \pm .070$** | $0.149 \pm .106$ | $0.073 \pm .072$ | $0.075 \pm .057$ |
| KL$(S \parallel R)$ ($\downarrow$) | TDDPM | **0.671** | **0.419** | **2.040** | **0.851** |
| | w/o local info | 0.900 | 2.198 | 2.432 | 2.234 |
| KL$(R \parallel S)$ ($\downarrow$) | TDDPM | **0.524** | **0.434** | **1.797** | **1.338** |
| | w/o local info | 0.965 | 1.573 | 2.147 | 1.746 |
| KL$_{sym}$ ($\downarrow$) | TDDPM | **0.597** | **0.426** | **1.919** | **1.095** |
| | w/o local info | 0.932 | 1.886 | 2.290 | 1.990 |
| JS ($\downarrow$) | TDDPM | **0.119** | **0.088** | **0.299** | **0.193** |
| | w/o local info | 0.179 | 0.312 | 0.351 | 0.320 |
| Density ($\downarrow$) | TDDPM | **0.037** | **0.029** | 0.197 | **0.036** |
| | w/o local info | 0.061 | 0.089 | **0.125** | 0.056 |
| Trip ($\downarrow$) | TDDPM | **0.070** | **0.042** | 0.203 | **0.047** |
| | w/o local info | 0.085 | 0.109 | **0.119** | 0.070 |
| Length ($\downarrow$) | TDDPM | **0.075** | **0.106** | **0.105** | **0.105** |
| | w/o local info | 0.086 | 0.114 | 0.171 | 0.303 |
| Pattern ($\uparrow$) | TDDPM | **0.900** | **0.860** | **0.780** | **0.840** |
| | w/o local info | 0.800 | 0.740 | 0.720 | 0.820 |

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

## A    APPENDIX / SUPPLEMENTAL MATERIAL

### A.1    EXTENDED RELATED WORK

TimeGAN (Yoon et al., 2019a) consists of a generative adversarial network (GAN) operating inside the latent space of an autoencoder. To further improve performance, they add an additional network with the task of predicting one time step ahead. The encoder, decoder, supervisor, generator and discriminator are all implemented using autoregressive models and in practice they use gated recurrent units (GRUs).

The TimeVAE (Desai et al., 2021) architecture is a variant of the popular variational autoencoder architecture. The autoencoder is trained with an additional loss component to have the latent space conform to a known statistical distribution, in this instance a multivariate normal distribution. The autoencoder is trained to both minimize the reconstruction loss, as well as minimizing the divergence between the embedded data and the prior set for the latent space.

COSCI-GAN (Seyfi et al., 2022) proposes to use a separate generative adversarial network for each channel of the data. The individual GANs all share single source of noise as input to the generator and, additionally they a central discriminator that is given the stacked output from the all the generators as input.

### A.2    ADDITIONAL ARCHITECTURE DETAILS

An overview of the approach is shown in Figure 3. Positional encoding (Ho et al., 2020):

$$\text{PE}_{(pos, 2i)} = \sin\left(-e^{i \frac{\log(10000)}{\frac{d}{2} - 1}}\right) \tag{4}$$

$$\text{PE}_{(pos, 2i+1)} = \cos\left(-e^{i \frac{\log(10000)}{\frac{d}{2} - 1}}\right) \tag{5}$$

The input to the transformer encoder is:

- L input tokens, each token corresponding to a time point in the noisy sequence. **Note:** The token size depends on the number of features of the dataset. It is a concatenation of:
  - $x \in \mathbb{R}^{DN}$, each corresponding to a dimension observed at each time point encoded using positional encoding
  - $x \in \mathbb{R}^{N}$, the time point encoded using the positional encoding introduced in (Vaswani et al., 2017)
  - $x \in \mathbb{R}^{N}$, a learned vector encoding denoting that this is a token that corresponds to a noisy sequence
- **Conditional information:** 64 tokens, each corresponding to a patch of the heatmap and being a concatenation of:
  - $x \in \mathbb{R}^{N}$, corresponding to the x position of the patch. Encoded using positional encoding (Vaswani et al., 2017).
  - $x \in \mathbb{R}^{N}$, corresponding to the y position of the patch. Encoded using positional encoding (Vaswani et al., 2017).
  - $x \in \mathbb{R}^{N}$, corresponding to the intensity of the heatmap. Encoded using a linear projection (Dosovitskiy et al., 2020).
  - $x \in \mathbb{R}^{N}$, a learned vector encoding denoting that this is a token that corresponds to the conditional information
- A token encoding the current denoising step:
  - $x \in \mathbb{R}^{N(D+1)}$, the denoising step encoded using positional encoding (Vaswani et al., 2017)

- $x \in \mathbb{R}^N$, a learned vector encoding denoting that this is a token that corresponds to the denoising step

We provide a list of hyperparameters in Table 5. During the pre-processing, we split trajectories into sub-trajectories if any observation leaves the geographic bounds, exceeds the velocity limit or if too long time has passed since the previous observation. The velocity limit is introduced to eliminate large velocities caused by sudden position changes. In the raw data, a trajectory can span several days with several hours between observations. Introducing a time-limit allows us to break these trajectories into individual and more time-constrained journeys. For the conditional dataset, we also calculate up to 10 regions for each trajectory. The region size is fixed and we set it to 1/10 of the width and height of the geographic bounds.

Table 5: Experiment hyperparameters

| Preprocessing | |
| --- | --- |
| Max velocity | 120 km/h |
| Max $\Delta t$ | 10 s |
| Sequence length | 128 |
| Region size | (0.01, 0.01) |
| Regions per trajectory | 10 |
| **Geographic Bounds** | |
| Geolife 100% | (39.9, 40, 116.3, 116.4) |
| Geolife 25% | (39.9, 39.95, 116.3, 116.35) |
| Porto 100% | (41.1, 41.2, -8.65, -8.55) |
| Porto 25% | (41.1, 41.15, -8.65, -8.6) |
| **Transformer** | |
| Hidden dimension $N$ | 64 |
| Number of heads | 4 |
| Number of layers | 2 |
| **Diffusion** | |
| Beta schedule | Cosine |
| Number of denoising steps | 100 |
| **Training** | |
| Batch size | 32 |
| Number of epochs | 100 |
| Optimizer | Adam |
| Learning rate | $2.9 \times 10^{-4}$ |
| $(\beta_1, \beta_2)$ | (0.9, 0.99) |

## A.3 ADDITIONAL RESULTS

### A.3.1 WHAT-IF ANALYSIS

Current approaches also struggle with environmental and context changes. The environment (and how people behave within it) often undergo rapid changes. This necessitates *data ageing mechanisms that allow models to adapt their prediction to changing circumstances* (Lana et al., 2018), such as for example road construction, traffic accidents, traffic-light malfunction or other traffic flow and road topology changes. For analysis and what-if scenario modeling, it is further important that the distributions of synthetic data is proportional to the probability distribution of the real data: Certain locations and motion patterns are more frequently occurring than others. This is important for policy making, planning and decisions to be resting on correct risk assessments using for example Bayesian inference.

Proof-of-concept of this capability by TDDPM is shown in figure 5.

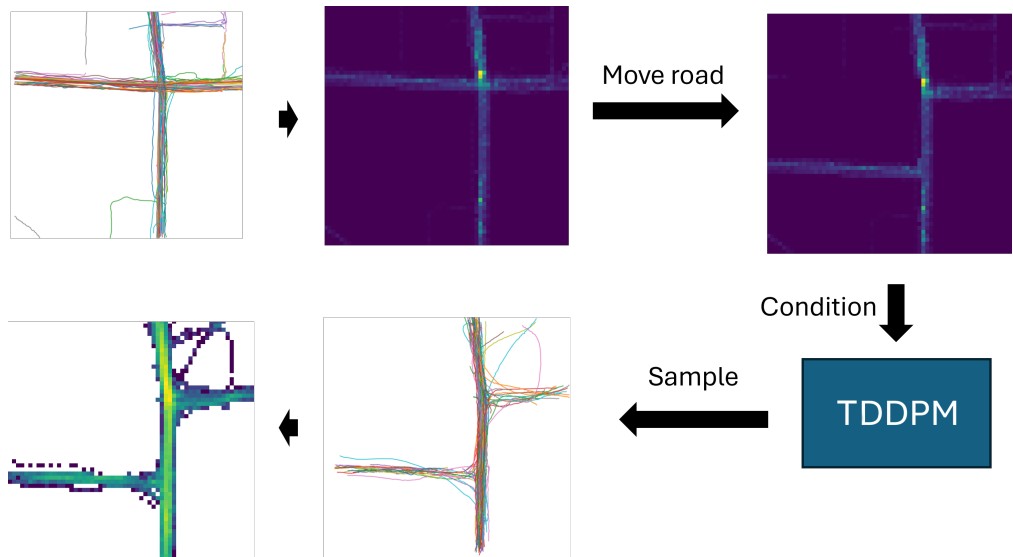

Figure 5: What-if-scenario where a road is removed and added.

### A.3.2 PRIVACY PRESERVING TRAJECTORY GENERATION

Here we elaborate on the privacy protection of our model and how it extends beyond the use of k-privacy.

We have two groups of users: 1) A set of users who have given consent to having their data used for training models (we chose bottom-left for ease of exposition) and 2) a set of users whose privacy we want to protect (remaining 3/4 of the data, again for ease of exposition). More specifically, we would like to protect against a membership inference attack where the trajectory of a single user could be extracted.

We use data from the first group to train the model, by extracting local information and training a conditional model, which maps from local information to individual trajectories. The trajectories of these users are not protected, as the model might memorize their trajectories.

For the second group we calculate local information, which is discretized spatially and time is marginalized out. The result is a heatmap, where each cell contains the number of observations in the corresponding geographical area. The private information left in the local information is already minimal. In the worst case scenario, if a single person were to travel in a less commonly traveled part of the map, a set of locations that the person visited could be extracted. The exact timing and in which order the areas were visited would still be unknown (protected by how the local information is calculated). For more visited areas, this risk would be minimal, as it would be difficult to isolate a single person's set of visited locations.

By applying k-privacy to the local information, we set cells with fewer than k observations to 0. This protects users (or groups of users) with fewer than k observations in each cell. Once applied, the risk of membership inference attacks is eliminated; an adversary cannot extract a sequence of locations a user visited (now even in less frequently visited areas).

### A.3.3 ADDITIONAL MEASURES

Here we investigate the quality of the resulting synthetic data following the evaluation methodology from DiffTraj (Zhu et al., 2023). The results are presented in Table 6. The measures are calculated for 3000 randomly drawn synthetic samples from each combination of model and dataset and 3000 randomly drawn samples from the training data. The measures are:

- **Density Error**: A pair of heatmaps of the training and synthetic data are calculated by dividing the city into $16 \times 16$ blocks. The number of observations in each block are counted and normalized. The Jensen-Shannon divergence of is between the training data heatmap and synthetic data heatmap.

- **Trip Error**: Two heatmaps pairs, each $16 \times 16$ blocks, are calculated. The first pair is of the start positions for all trajectories, one heatmap for the training data and another for the synthetic data. The second pair is calculated from the last position for all trajectories. The Jensen-Shannon divergence is calculated once for each pair and then the average is reported.

- **Length Error**: The distance between consecutive observations are calculated, once for the training data and once for the synthetic data. Histograms are calculated of each, with the number of bins set to 16. Finally, the Jensen-Shannon divergence is calculated from the histogram of training data to the histogram of the synthetic data.

- **Pattern Score**: Using the heatmaps from Density Error, the top N areas (highest count) from the training and synthetic data are collected. The F-score is then calculated and reported.

Table 6: Evaluation measures based on the evaluation done in DiffTraj (Zhu et al., 2023).

| Dataset | Model | Density ($\downarrow$) | Trip ($\downarrow$) | Length ($\downarrow$) | Pattern ($\uparrow$) |
|---|---|---|---|---|---|
| Geolife 25% | TimeGAN | 0.320 | 0.362 | 0.027 | 0.500 |
| | TimeVAE | 0.141 | 0.171 | 0.459 | 0.630 |
| | COSCI-GAN | 0.197 | 0.249 | 0.579 | 0.540 |
| | Diffusion-TS | 0.058 | 0.086 | **0.014** | 0.890 |
| | DiffTraj | 0.229 | 0.246 | 0.231 | 0.570 |
| | TDDPM | **0.037** | **0.070** | 0.075 | **0.900** |
| Geolife 100% | TimeGAN | 0.201 | 0.238 | **0.013** | 0.700 |
| | TimeVAE | 0.060 | 0.081 | 0.072 | 0.760 |
| | COSCI-GAN | 0.139 | 0.166 | 0.729 | 0.680 |
| | Diffusion-TS | 0.061 | 0.076 | 0.020 | 0.820 |
| | DiffTraj | 0.175 | 0.185 | 0.301 | 0.730 |
| | TDDPM | **0.029** | **0.042** | 0.106 | **0.860** |
| Porto 25% | TimeGAN | 0.163 | 0.166 | 0.075 | 0.710 |
| | TimeVAE | 0.073 | 0.117 | 0.349 | 0.740 |
| | COSCI-GAN | 0.116 | 0.193 | 0.677 | 0.660 |
| | Diffusion-TS | **0.061** | **0.076** | **0.014** | 0.760 |
| | DiffTraj | 0.172 | 0.169 | 0.459 | 0.620 |
| | TDDPM | 0.197 | 0.203 | 0.105 | **0.780** |
| Porto 100% | TimeGAN | 0.132 | 0.154 | 0.017 | 0.760 |
| | TimeVAE | 0.051 | 0.080 | 0.201 | **0.840** |
| | COSCI-GAN | 0.102 | 0.166 | 0.612 | 0.780 |
| | Diffusion-TS | 0.056 | 0.068 | **0.010** | 0.830 |
| | DiffTraj | 0.406 | 0.363 | 0.437 | 0.420 |
| | TDDPM | **0.036** | **0.047** | 0.105 | **0.840** |

### A.3.4 THE EFFECT OF TRAINING DATA SELECTION ON GENERALIZATION

In this experiment, we investigate the impact of the choice of geographical data on generalization. We train TDDPM on two different sub-regions, the bottom left quadrant of Geolife 100% and the top right quadrant. We then compare to all other models, including TDDPM, trained on data from all of Geolife 100%. The results are presented in Table 7. We see that TDDPM trained on top-right quadrant performs best on the TSTR measure, giving our approach the best score out of all models we have compared. We also see that training on the bottom-left yields an improvement over using training data from all of Geolife 100%, at the cost of slightly higher $\text{KL}(S||R)$. This suggests that data from some geographical areas might contribute more towards certain scores or aspects of performance. Further, if

the intended use of the synthetic data is known, simply selecting all available training data might not give the best possible synthetic data. This opens up a promising direction for future work: For a given task, from what regions should the training data be gathered such that the resulting synthetic data is the most useful?

Table 7: Results of experiment evaluating the effect of geographical area on generalization performance. This experiment evaluates the performance of generating a synthetic dataset for Geolife 100%. There are three variants of TDDPM in this experiment, each having access to different data. The models that does not have access to all data are generalizing to the remaining geographical area. **\*** For additional clarity, we mark the models that are trained on sub-regions and are generalizing to the entire region.

| Model | Data | TSTR | $KL(S \parallel R)$ | $KL(R \parallel S)$ | $KL_{sym}$ | JS |
|---|---|---|---|---|---|---|
| TimeGAN | | $0.107 \pm 0.096$ | 2.712 | 2.175 | 2.443 | 0.378 |
| TimeVAE | | $0.089 \pm 0.084$ | 2.419 | 1.821 | 2.120 | 0.337 |
| COSCI-GAN | All | $0.090 \pm 0.090$ | 2.546 | 1.922 | 2.234 | 0.370 |
| Diffusion-TS | | $0.102 \pm 0.097$ | 2.048 | 1.542 | 1.795 | 0.298 |
| DiffTraj | | $0.092 \pm 0.078$ | 2.909 | 2.287 | 2.598 | 0.393 |
| TDDPM | | $0.127 \pm 0.089$ | **0.419** | 0.434 | 0.426 | 0.088 |
| TDDPM**\*** | Bottom-left | $0.094 \pm 0.076$ | 0.453 | **0.378** | **0.416** | **0.085** |
| TDDPM**\*** | Top-right | $\mathbf{0.086 \pm 0.086}$ | 0.538 | 0.453 | 0.496 | 0.100 |

### A.3.5 ADDITIONAL VISUALIZATION

Figure 6 show a detailed view of Geolife 25%, including training data, a heatmap of the training data, the synthetic data from TDDPM as well as a heatmap of the synthetic data. In Figure 7 we show TDDPM trained on Geolife 25% generalizes to Geolife (100%) without re-training. By comparing the samples, we see that the synthetic samples are similar to the original samples. The heatmaps are also similar, meaning that the model learns to preserve proportionality.

We also show random samples for random regions of the various datasets, comparing random samples from the original data to the synthetic data from the various models. This is shown in Figures 8, 9, 10 and 11. TDDPM, our proposed method, produces the synthetic samples that are most similar to the real data. The main exception is Porto 100% where it struggles to generate data that is similar to the real data. Diffusion-TS comes in as second, which can generate data for Geolife 25% but struggles to scale up to GeoLife 100% and the samples for Porto. The trajectories from the other methods show little to no resemblance to the real data.

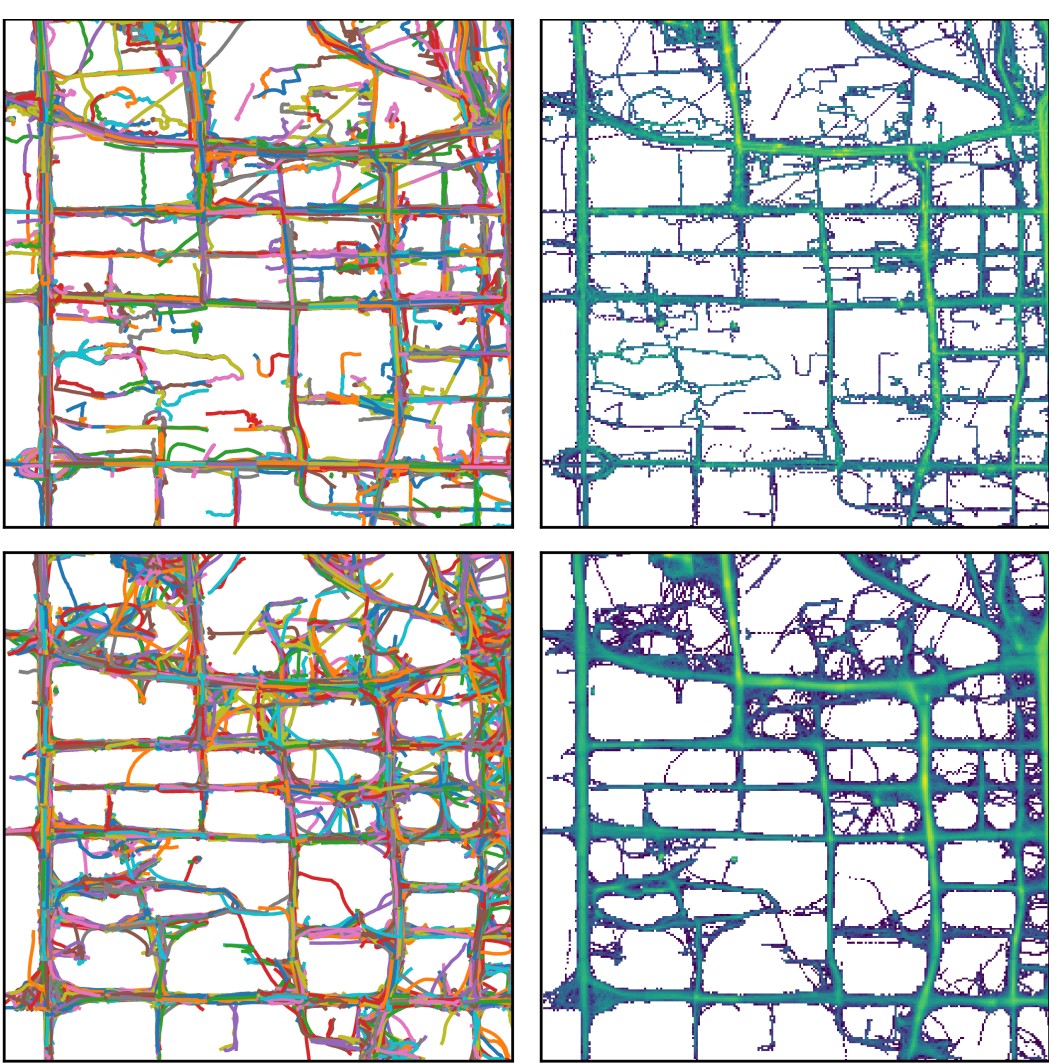

Figure 6: Interpolation experiment. The model has trained on this region is tasked to reconstruct it from the heatmaps. *Top left:* Data from Geolife used for training. *Top right:* heatmap of training data and areas used for creating query heatmaps for sampling the model, *Bottom left:* synthetic trajectories. *Bottom right:* heatmap of the synthetic data.

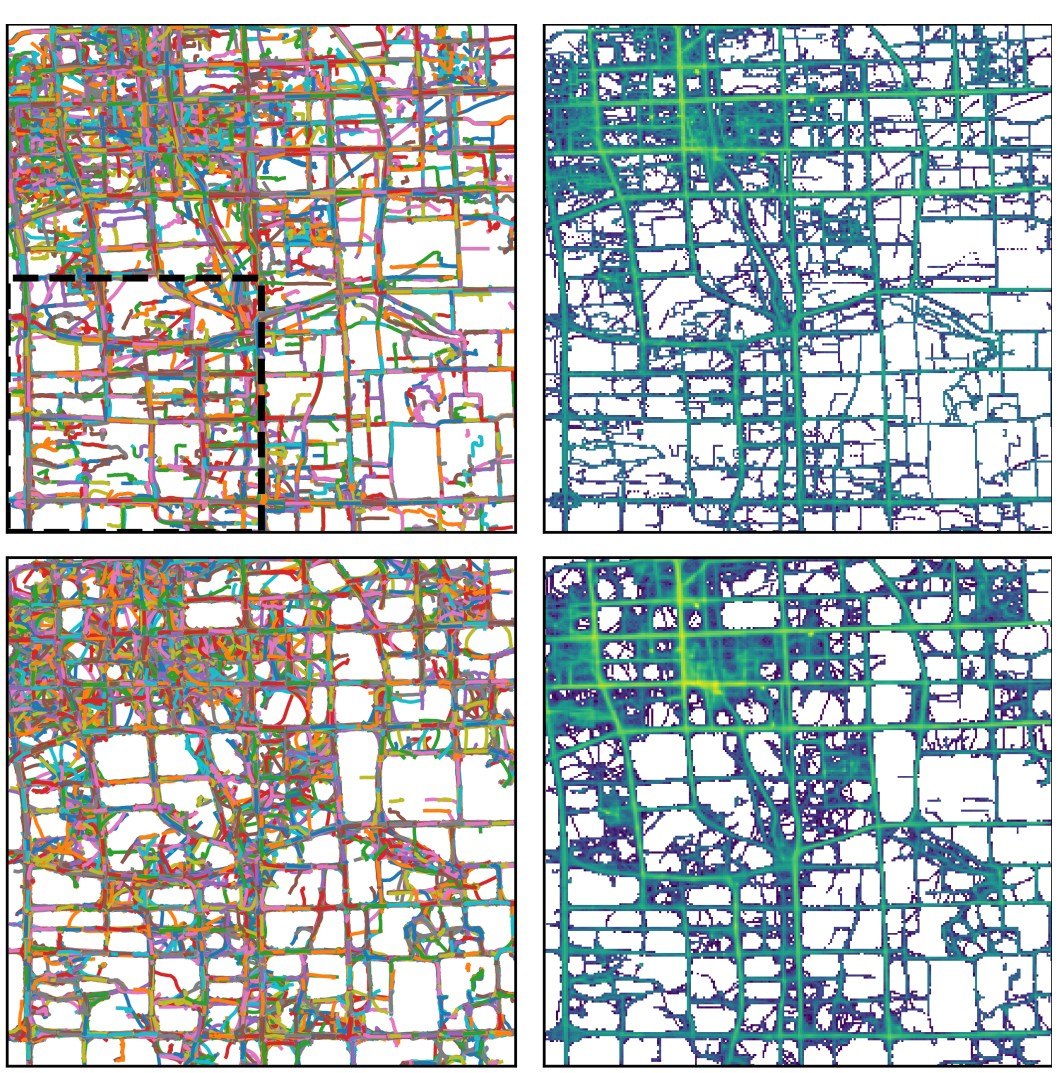

Figure 7: Generalization experiment. The model is trained on the lower left quadrant and used to generate data on the remaining geographical area. *Top left:* Data from Geolife, the lower quandrant of which used for training. *Top right:* heatmap of training data, *Bottom left:* synthetic trajectories. *Bottom right:* heatmap of the synthetic data.

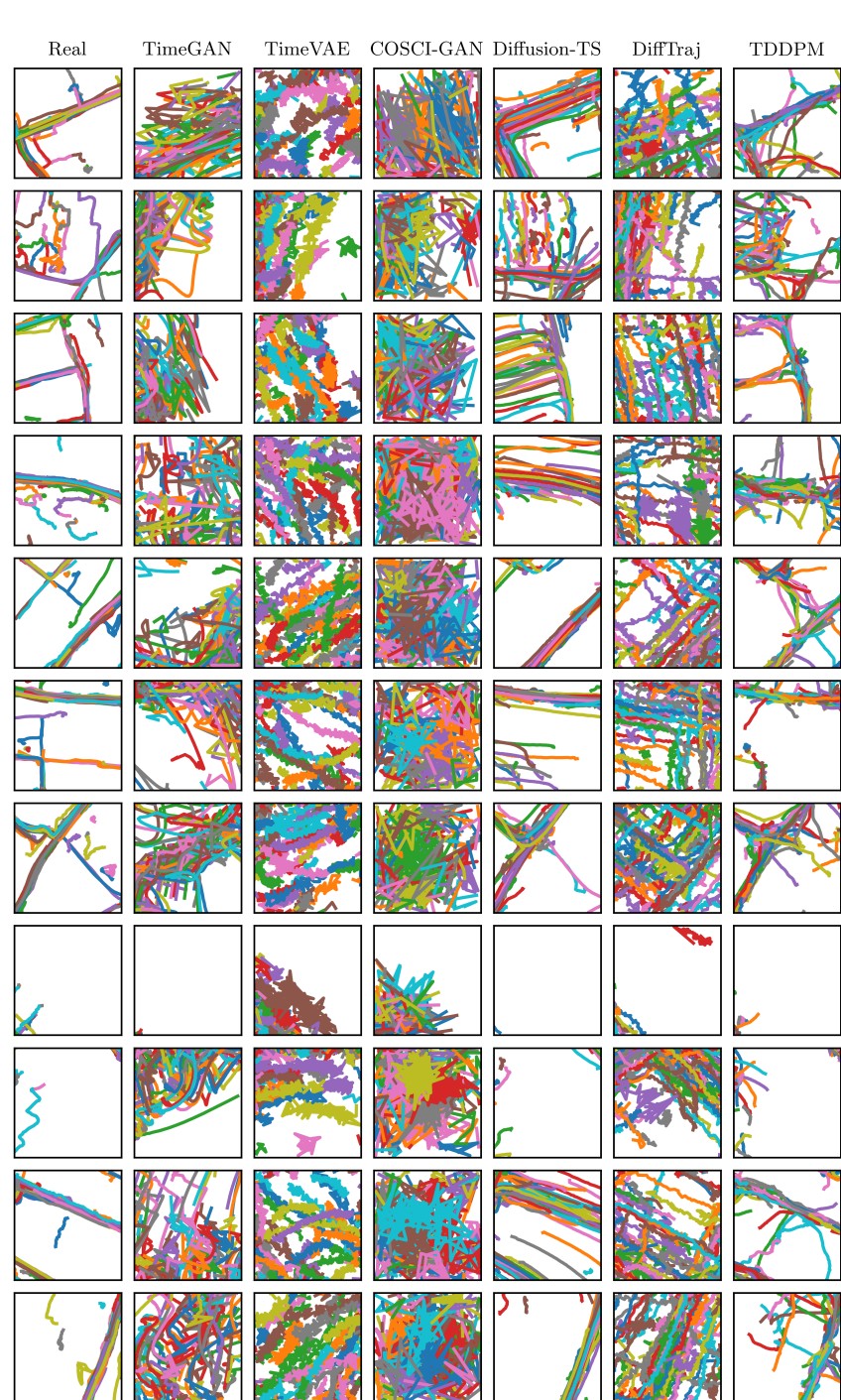

Figure 8: Random samples from training data and synthetic data across 11 different regions chosen at random, all from Geolife 25%

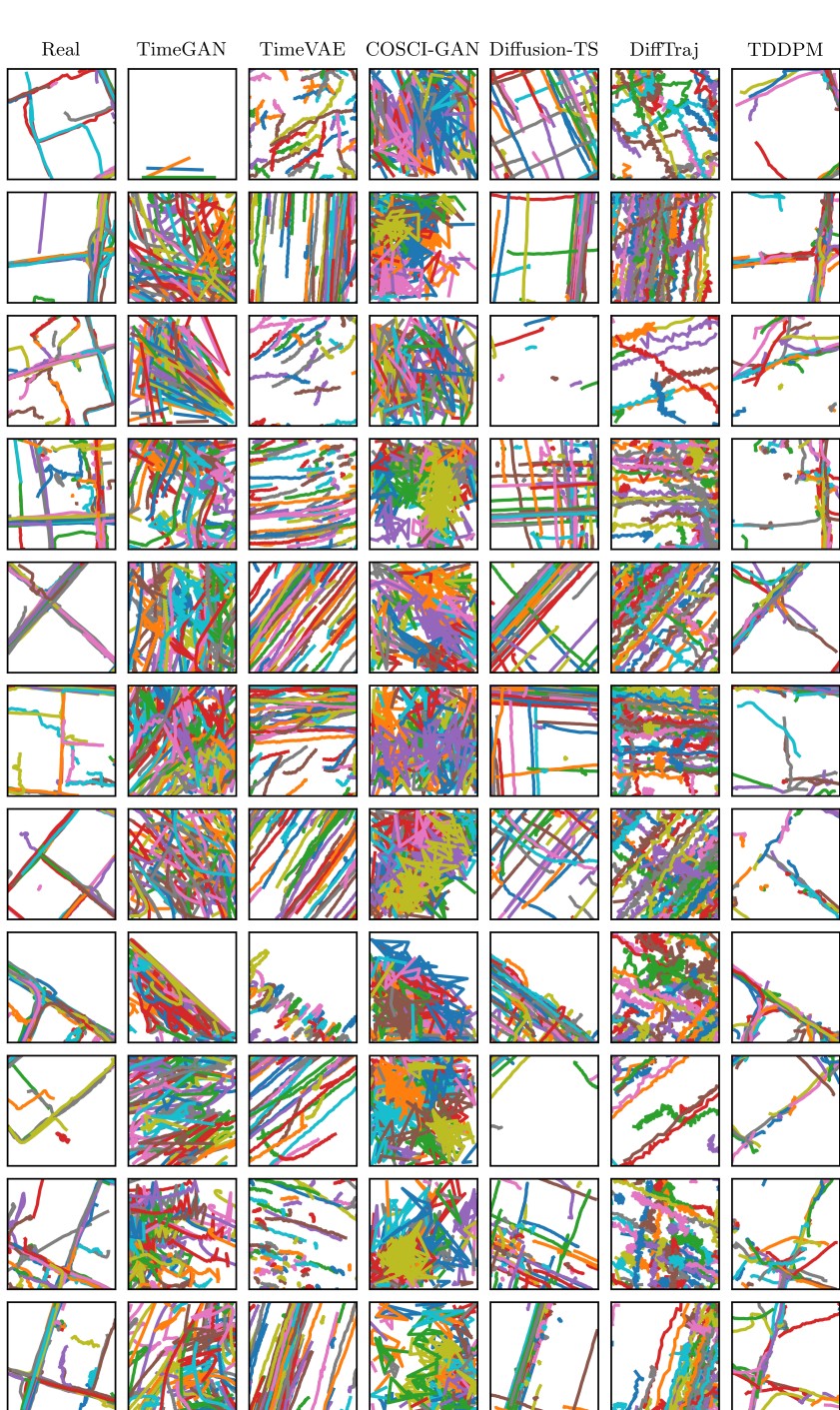

Figure 9: Random samples from training data and synthetic data across 11 different regions chosen at random, all from Geolife 100%

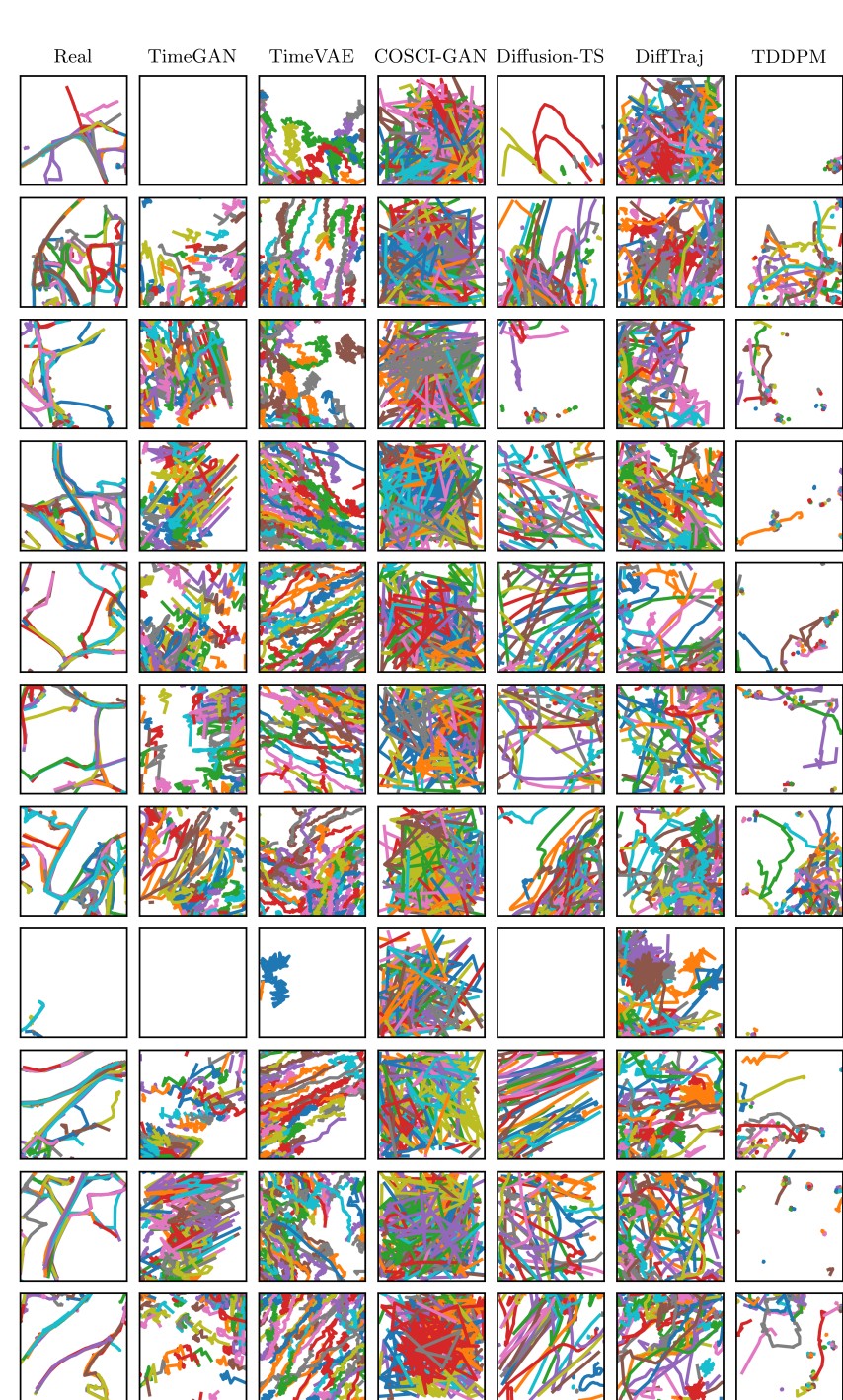

Figure 10: Random samples from training data and synthetic data across 11 different regions chosen at random, all from Porto 25%

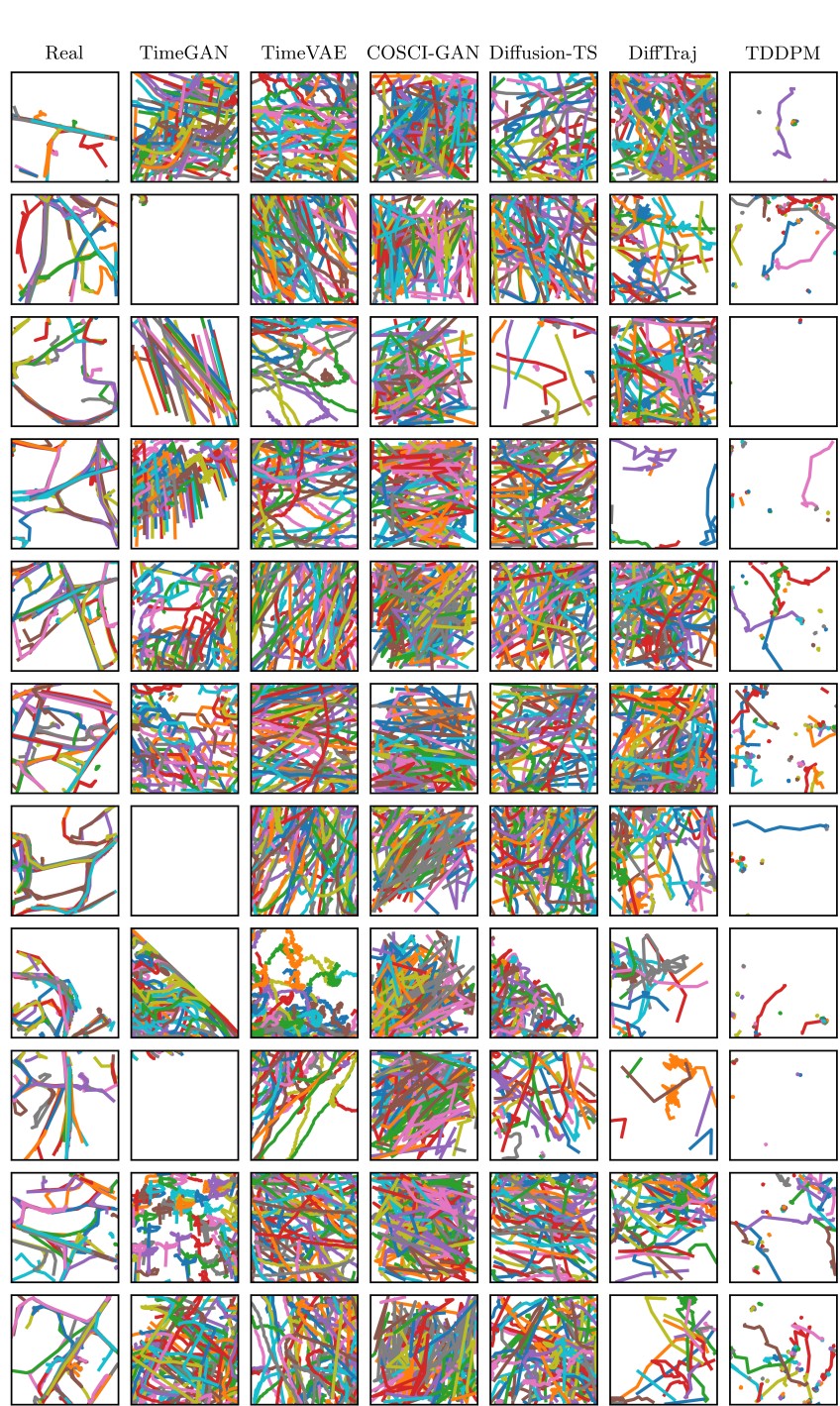

Figure 11: Random samples from training data and synthetic data across 11 different regions chosen at random, all from Porto 100%

