# OpenReview forum: "Deep Temporal Deaggregation: Large-Scale Spatio-Temporal Generative Models"
_ICLR.cc/2025/Conference — Submitted to ICLR 2025_

### Official Review · Reviewer_3LhK · 2024-11-02

**Soundness:** 2
**Presentation:** 1
**Contribution:** 2
**Rating:** 3
**Confidence:** 4

**Summary:**

This paper propose TDDPM, a spatiotemporal generative model for trajectories. It aims to address two issues, (1) shortage of publicly available trajectories data; (2) model should conduct predictions about unobserved parts in space. To address these issues, TDDPM applies occupancy frequency marginal distribution as local information and hierarchical occupancy frequency mixture. The authors demonstrate the effectiveness of TDDPM on real-world datasets, showing improvements in forecasting accuracy compared to baseline models.

**Strengths:**

1. The paper focuses on the privacy issues of spatio-temporal trajectory data, as well as enhancing the prediction about unobserved parts in space. This is a very valuable and meaningful research topic.
2. The paper provides detailed visualizations and What-if analysis.

**Weaknesses:**

1. This paper proposes a privacy-preserving for generating synthetic trajectory samples, however, its contribution to privacy protection is limited. The paper uses only k-anonymity to protect local information in Section 5.3, which is a straightforward design.
2. There are no baselines in Section 5.2 and Section 5.3. The author should demonstrate how other methods perform in the Generalization experiment and with k-anonymity.

**Questions:**

Could you provide brief descriptions of your baselines?

---

> ### Author Response · Authors · 2024-11-21
>
> We would like to thank the reviewer for their feedback and time. We have addressed the concerns below:
>
> > This paper proposes a privacy-preserving for generating synthetic trajectory samples, however, its contribution to privacy protection is limited. The paper uses only k-anonymity to protect local information in Section 5.3, which is a straightforward design.
>
> We would like to clarify that this paper proposes a new approach to synthetic data generation for trajectories, which has several applications. Among these are data augmentation (generating more data but with similar statistics), data compression (a more compact representation of a data set), what-if-analysis (generate data about hypothetical scenarios not explicitly observed in the training data) and sharing useful but non-sensitive versions of sensitive data. The latter, if the sensitive data is connected to privacy, is the task of privacy-preserving synthetic data generation of  trajectory samples. We also want to point out that the task of unconditional data generation is distinctly different from forecasting and (point) prediction, but is a foundation which can be adapted for forecasting and prediction tasks [1].
>
> While we agree that k-anonymity is relatively straight-forward, the design that allows for its application is non-trivial. Previous trajectory generation approaches have failed to combine both privacy-preservation and utility [2] on the in-distribution generalization task  of unconditional synthetic data generation. In this work we propose the first method (to our knowledge) that achieves out-of-distribution generalization for this task. We demonstrate how this novel capability can achieve privacy (k-anonymity) with minimal modification, showcasing both a strength and a concrete application of our approach.
>
> [1] Alcaraz, J. L., & Strodthoff, N. (2023). Diffusion-based Time Series Imputation and Forecasting with Structured State Space Models. Transactions on Machine Learning Research.
>
> [2] Buchholz, E., Abuadbba, A., Wang, S., Nepal, S., & Kanhere, S. S. (2024). SoK: Can Trajectory Generation Combine Privacy and Utility?. Proceedings on Privacy Enhancing Technologies, 3, 1-19.
>
> > There are no baselines in Section 5.2 and Section 5.3. The author should demonstrate how other methods perform in the Generalization experiment and with k-anonymity.
>
> Since there are no other methods capable of out-of-distribution generalization for synthetic trajectory data generation (to the best of our knowledge), we are only comparing the quality of the k-privacy synthetic data to synthetic data from the version of our model that has been trained on all the data. The latter acts as a baseline on how good performance can be achieved using all data, but it is inherently not privacy-preserving.
>
> The proposed k-privacy synthetic data generation model has not trained on data from the regions it is generating synthetic data from, so it is impossible for the model to memorize any data there, and it is only given aggregate statistics over a set of trajectories (k in k-privacy) as conditional, which together means that strong k-privacy is achieved.
>
> An interesting path for future work is to investigate differential privacy algorithms for calculating local information [3], as this could allow us to increase utility while maintaining similar levels of privacy.
>
> [3] Ghazi, Badih, et al. "Differentially private heatmaps." Proceedings of the Thirty-Seventh AAAI Conference on Artificial Intelligence and Thirty-Fifth Conference on Innovative Applications of Artificial Intelligence and Thirteenth Symposium on Educational Advances in Artificial Intelligence. 2023.
>
> > Questions: Could you provide brief descriptions of your baselines?
>
> We have provided a brief description of DiffTraj and Diffusion-TS in the Related Work section (rows 113-123), and of TimeGAN, TimeVAE and COSCI-GAN in Appendix A.1 (rows 598 - 612). Could you clarify your question?

---

> > ### Comment · Reviewer_3LhK · 2024-11-26
> > **Acknowledge**
> >
> > This message acknowledges the authors' response. Regarding the first question, I think the diverged claim of research contribution endorses the quality concern of this paper. The claim of privacy-preservation seems to be added at the last minute before submission. The method used for privacy preservation is straightforward and the main body lacks discussion about the issue of privacy protection. Regarding the second question, it seems straight forward to test the performance of baselines in unseen environments with k-anonymity. Considering both concerns are not well addressed, I would like to maintain my score.

---

> > > ### Author Response · Authors · 2024-11-27
> > >
> > > > I think the diverged claim of research contribution endorses the quality concern of this paper.
> > >
> > > The intent with listing well-recognized applications for (unconditional) synthetic data generation for trajectories (what the paper is about) is to show that there are other applications than privacy-preserving synthetic data generation. We do not claim that these are our contributions. The main contribution of the paper is the significant improvement upon state-of-the-art for unconditional synthetic data generation for trajectories. Other contributions are the two applications, where we demonstrate how TDDPM, by its design, is capable of what-if-analysis and privacy-preserving synthetic data generation. That the applications are easy to realize is a consequence of how the TDDPM method works, which is a major strength of the method.
> > >
> > > We agree that a paper focusing on privacy should have more thorough experiments substantiating the contributions to the privacy field. The focus of the paper is however NOT on privacy-preserving synthetic data generation, it is on high-quality unconditional synthetic data generation, with an approach that is compatible with what-if-analysis and certain kinds of privacy. We acknowledge that the use of privacy (and protection for other sensitive attributes) as a motivator for why synthetic data generation is important, from abstract and introduction, can cause confusion on the subject. We will update the text to make this more clear.
> > >
> > > > The method used for privacy preservation is straightforward and the main body lacks discussion about the issue of privacy protection.
> > >
> > > We can elaborate on the privacy protection of our model and how it extends beyond the use of k-privacy:
> > > We have two groups of users: 1) A set of users who have given consent to having their data used for training models (we chose bottom-left for ease of exposition) and 2) a set of users whose privacy we want to protect (remaining 3/4 of the data, again for ease of exposition). More specifically, we would like to protect against a membership inference attack where the trajectory of a single user could be extracted.
> > >
> > > We use data from the first group to train the model, by extracting local information and training a conditional model, which maps from local information to individual trajectories. The trajectories of these users are not protected, as the model might memorize their trajectories.
> > >
> > > For the second group we calculate local information, which is discretized spatially and time is marginalized out. The result is a heatmap, where each cell contains the number of observations in the corresponding geographical area. The private information left in the local information is already minimal. In the worst case scenario, if a single person were to travel in a less commonly traveled part of the map, a set of locations that the person visited could be extracted. The exact timing and in which order the areas were visited would still be unknown (protected by how the local information is calculated). For more visited areas, this risk would be minimal, as it would be difficult to isolate a single person’s set of visited locations.
> > >
> > > By applying k-privacy to the local information, we set cells with fewer than k observations to 0. This protects users (or groups of users) with fewer than k observations in each cell. Once applied, the risk of membership inference attacks is eliminated; an adversary cannot extract a sequence of locations a user visited (now even in less frequently visited areas).
> > >
> > > > Regarding the second question, it seems straight forward to test the performance of baselines in unseen environments with k-anonymity.
> > >
> > > We do not understand your request. To the best of our knowledge, there is no straightforward way to apply k-anonymity to either the datasets directly or to the baseline models we use in our paper. We might be missing something on our side, or there is a misunderstanding. Can you elaborate on how we can test the performance of the baselines in unseen environments with k-anonymity?
> > >
> > > Secondly, we do not understand how we can compare with other baselines in unseen environments. We are not aware of any baselines that are able to do out-of-distribution generalization. Can you clarify what you think is missing from the paper, so that we may address it?
> > >
> > > As we stated in our previous response, due to a lack of available baselines with the same capabilities as our method, the best we can do is to use models that do not use k-anonymity and do not generalize to unseen environments as baselines. This means that our method is compared to baselines with full access to all data, including information treated as private for our model. We use the same evaluation measures in these experiments as we do in the large comparison in Table 1, meaning that they are directly comparable and form the baseline for our other experiments.

---

> > > > ### Author Response · Authors · 2024-11-29
> > > >
> > > > Thank you for the constructive feedback!
> > > >
> > > > Upon closer inspection on what we actually say about privacy, we noticed that the pdf-file had the remainder of a previous working title. A title that posed the paper to have a focus on privacy. It is an embarrassing mistake that we take full responsibility for. We apologize for this and have now corrected the pdf-title to correspond to the correct title, as it is in the open review submission.
> > > > We have now clarified what our main contributions are as an explicit list at the end of the introduction, where we also make it clear how our contributions relate to privacy as one possible application we consider.
> > > >
> > > > We furthermore clarified in Section 5.2 and Section 5.3 that Table 1 contains the baseline results for the generalization experiment and the privacy experiment, respectively. This is now properly referenced in the text in both sections. We hope that this addresses the concern of a lack of baselines for unseen environments and k-anonymity experiments: “Regarding the second question, it seems straight forward to test the performance of baselines in unseen environments with k-anonymity”.
> > > >
> > > > We have also added a discussion on privacy protection in the appendix (A.3.2), referred to from the Privacy experiment section (Section 5.3). We can move more of this to the main text in later revisions.

---

### Official Review · Reviewer_jUbV · 2024-11-02

**Soundness:** 3
**Presentation:** 3
**Contribution:** 2
**Rating:** 6
**Confidence:** 2

**Summary:**

This paper presents TDDPM, a model for generating high-fidelity, privacy-preserving trajectory data in complex environments. TDDPM leverages a denoising diffusion approach to deaggregate spatial data into individual trajectories, allowing realistic time-series generation that generalizes to unseen areas. By conditioning on spatial aggregates, it achieves strong out-of-distribution performance. The model also uses k-anonymity for privacy and introduces a new benchmark for evaluating synthetic trajectory data, making it a valuable tool for urban planning and autonomous driving applications.

**Strengths:**

(1) The demonstration of proposed method TDDPM is detailed.
(2) The authors describe the motivation and background of generating out-of-distribution trajectories in detail.

**Weaknesses:**

1. The contributions of TDDPM are unclear. The authors should clearly claim the contributions in the end of introduction.
2. In Table 1 and Table 2, standard deviation of KL and JS divergence are not reported.
3. This paper lacks an ablation study part. Experiments should be added to verify the effectiveness of proposed two steps mentioned in Section 4.
4. The experiment setting is confusing. The authors claim that TDDPM could achieve out-of-distribution generalization in Abstract. More analysis should be added to demonstrate the diffirence between the synthetic dataset and Geolife/Porto.
5. Line 308 to line 325 seems making no sense.

**Questions:**

See weaknesses.

---

> ### Author Response · Authors · 2024-11-21
>
> We would like to thank the reviewer for their feedback and time. We have addressed the concerns below:
>
> > The contributions of TDDPM are unclear. The authors should clearly claim the contributions in the end of introduction.
>
> Thank you for the suggestion to increase the clarity of our contributions. We will update the paper with clear claims of contributions at the end of the introduction.
>
> > In Table 1 and Table 2, standard deviation of KL and JS divergence are not reported.
>
> We understand that this is done when KL or similar distribution measures are estimated (e.g. when not possible to calculate using all available data). However, in this work we can directly calculate these measures from the estimated distributions, and as such the measures are exact and not estimates.
>
> Can you clarify what you would like to know, and if possible point us to any related work on generative models where they use KL or similar distribution measures and where they do report standard deviation even though they can directly calculate the measure?
>
> > This paper lacks an ablation study part. Experiments should be added to verify the effectiveness of proposed two steps mentioned in Section 4.
>
> Thank you for the valuable suggestion. We have now added an ablation study A.3.3 (Appendix) where we compare the two-step unconditional trajectory generation approach (with internal conditioning) with a corresponding single-step trajectory generation approach (without internal conditioning).
>
> > The experiment setting is confusing. The authors claim that TDDPM could achieve out-of-distribution generalization in Abstract. More analysis should be added to demonstrate the difference between the synthetic dataset and Geolife/Porto.
>
> We have supplemented the result with additional evaluation as per the request of reviewer URHw, this adds further comparison between the real and synthetic data and is presented in A.3.2. We have also shown more examples of out-of-distribution generalization as part of our experiment where we compare the generalization performance on two instances of our model, each having trained on different datasets. This is shown in A.3.4.
>
> > Line 308 to line 325 seems making no sense.
>
> Thank you for identifying this opportunity to increase the clarity of our presentation. There are several perspectives on synthetic data quality in the literature, but no concise description that span all of these. A contribution of this paper is that we harmonize and combine the perspectives from [Esteban, Wu and Alaa] into a coherent and more complete quality model. Based on this, we ground our evaluation measures in the different quality notions.
>
> We will update the text to make this clearer.

---

> > ### Comment · Reviewer_jUbV · 2024-11-25
> >
> > The additional experiments in the Appendix do address most of my concerns, so I will increase my score. In addition, some experiments in the Appendix should be placed in the main text, especially the ablation study part. I suggest that the authors should optimize the content of the main text to make the experimental part more substantial.

---

> > > ### Author Response · Authors · 2024-11-29
> > >
> > > > The additional experiments in the Appendix do address most of my concerns, so I will increase my score. In addition, some experiments in the Appendix should be placed in the main text, especially the ablation study part. I suggest that the authors should optimize the content of the main text to make the experimental part more substantial.
> > >
> > > Thank you for the constructive feedback! We have now moved the following experiments from the appendix to the main text: additional evaluation using measures from DiffTraj (merged into Table 1) and the ablation study (Section 5.3.1). is now a separate experiment after the k-privacy experiment. (Note that we accidentally placed it at the wrong level, it should be 5.4 not 5.3.1).
> > >
> > > As per your previous comment, we have updated the text previously between lines 308 and 325 (now located on lines 325 to 329 due to said list of contributions) to more clearly explain our intent of harmonizing several qualities of synthetic data. We have also clarified the contributions of our work by adding a list of contributions to the end of the introduction.

---

### Official Review · Reviewer_URHw · 2024-11-03

**Soundness:** 2
**Presentation:** 3
**Contribution:** 3
**Rating:** 6
**Confidence:** 4

**Summary:**

This paper introduces a new task—out-of-distribution generalization—for synthetic trajectory generation, which current models in the literature cannot address. To enable this generalization capability, the paper proposes using a heatmap as a conditioning constraint in a generative denoising diffusion model. Experiments on both unconditional trajectory generation and generation for new environments and hypothetical scenarios are conducted to evaluate its effectiveness. Additionally, the generation performance is analyzed under private heatmap scenarios.

**Strengths:**

1. This paper addresses a novel task, enabling the synthetic trajectory generation model to generalize to new areas.
2. A heatmap is used as a novel conditioning mechanism to achieve this generalization capability in the trajectory generation model.
3. Extensive experiments are conducted to evaluate the proposal.

**Weaknesses:**

1. Additional metrics, such as Density Error, Trip Error, Length Error, and Pattern Score (as used in the TrajDiff paper), should be considered for evaluation.
2. The proposed method splits regions for training, but it is unclear whether the model can correctly generate cross-region trajectories under these conditions.
3. In Table 2, the reasons why the model trained on 25% outperforms the model trained on 100% in most cases should be explained more clearly.
4. As an ablation study, it would be helpful to include results when the model does not use the conditioning component for the unconditional trajectory generation task.
5. Providing information on hyperparameters and including a complexity analysis would improve the paper.
6. Typo:  "ell(..." should be "l(..." on Page 5.

**Questions:**

Same as the weaknesses

---

> ### Author Response · Authors · 2024-11-21
>
> We would like to thank the reviewer for their feedback and time. We have addressed the concerns below:
>
> >Additional metrics, such as Density Error, Trip Error, Length Error, and Pattern Score (as used in the TrajDiff paper), should be considered for evaluation.
>
> We have implemented the measures from the DiffTraj paper and present the results from using them in A.3.2 (appendix).
>
> >The proposed method splits regions for training, but it is unclear whether the model can correctly generate cross-region trajectories under these conditions.
>
> The model can correctly generate cross-region trajectories: The regions used during training and testing are distinctly different, and these regions do partially overlap each other. The training ones are randomized with respect to translation and rotation, while at test time the regions are cells on a rigid grid. That is, a test-time region (inside the support of the training data) does partially overlap multiple training regions. The model can correctly generate trajectories on the rigid grid, as can be seen in for example Figure 8, A3.5 (Appendix).
> However, the model has other limitations. For example, it cannot generate trajectories that span multiple disjoint test regions, a limitation that suggests a promising avenue for future work.
>
> > In Table 2, the reasons why the model trained on 25% outperforms the model trained on 100% in most cases should be explained more clearly.
>
> We suspect that this is due to increased positioning noise in the densely populated area with high buildings (shadowing, multi-pathing, jumps to WiFi AP geolocation, among others.). The area in the lower left quarter is less susceptible to noise and by training on data from only this area, we learn a low-noise model which is a better fit for the entirety of the map. If true, this opens up an interesting direction for future work: Is there an area which when trained on gives better generalization performance? We will update the conclusion to address this possible direction.
>
> We have updated the paper with an experiment where we train our model on three different datasets: one on the top-right quarter of the Geolife dataset, another on the bottom-left quarter and another trained on data from the entire region. The evaluation of these models are available in A.3.4 (Appendix). Depending on which evaluation measure is being used to determine the best model, different instances of TDDPM are achieving the best performance. This suggests that using all available data might not be the best, instead a subset of all available data that can yields the best performance. Further, this subset will probably depend on the downstream task.
>
> > As an ablation study, it would be helpful to include results when the model does not use the conditioning component for the unconditional trajectory generation task.
>
> Yes, thank you for the suggestion. We have now added an ablation study in A.3.3 (appendix, Table 6),  where we compare the two-step unconditional trajectory generation approach (with internal usage of a conditioning component) with a corresponding single-step trajectory generation approach (without a conditioning component). For completeness, we also include the evaluation measures from DiffTraj as suggested.
>
> > Providing information on hyperparameters
>
> We now have a list of hyperparameters in A.2.
>
> > and including a complexity analysis would improve the paper.
>
> Could you clarify and elaborate on what you would like the complexity analysis to cover? Providing a complexity analysis is not common in the field of unconditional time-series generative models, so that is why we ask. In general, we share time and memory-complexity with the transformer model that we use for denoising in the denoising diffusion model. The transformer scales quadratically, both in terms of time and memory consumption, with the size of the input. In our case, the input consists of local information (64 tokens fixed), the denoising step (1 token) and a sequence to be denoised (1 token per time step). If we were to increase our models to longer sequences or to larger areas, it would come at the cost of quadratic time and memory. With that said, time and memory complexity is not the focus of this work. If it were, we would start by looking into the more memory efficient variants of the transformer architecture, e.g. Informer or Perceiver to reduce complexity.
>
> Quantitatively, it takes 65 minutes to train our model on Geolife 25% where the local information is pre-computed. Once trained, it takes 15 minutes to draw 10 000 samples for Geolife 100 with TDDPM (each a trajectory with sequence-length 128) and this includes calculating the local information for each region. In the ablation without local information, we measure 5 minutes for training and 69 seconds to sample the model. All experiments are run on a machine with an AMD 5900X, 128 Gb of RAM and a single RTX 3090 Ti.
>
> > Typo: “ell(…” should be “l(…” on Page 5.
>
> Thanks for pointing that out!

---

> ### Comment · Reviewer_URHw · 2024-11-25
>
> Thank you very much for your responses. I have updated my score. However, in my opinion, generating trajectories that span multiple disjoint test regions remains a significant issue for the current proposal. This might be also a potential reason for the low performance in Length Error. In addition, the complexity, including time and memory consumption, should be compared with baselines.

---

> > ### Author Response · Authors · 2024-11-29
> >
> > > Thank you very much for your responses. I have updated my score. However, in my opinion, generating trajectories that span multiple disjoint test regions remains a significant issue for the current proposal. This might be also a potential reason for the low performance in Length Error. In addition, the complexity, including time and memory consumption, should be compared with baselines.
> >
> > Thank you for the constructive feedback!
> >
> > Since our last update, we have moved the results for the evaluation using measures Density, Trip, Length and Pattern from the appendix to Table 1 in the main text. We have also moved the ablation study from the appendix to the main text.

---

### Comment · Area_Chair_mk6M · 2024-11-25
**Acknowledge the author responses**

Dear Reviewers,

Thank you very much for your effort. As the discussion period is coming to an end, please acknowledge the author responses and adjust the rating if necessary.

Sincerely,
AC

---

### Author Response · Authors · 2024-11-29

We would like to thank all reviewers for their constructive feedback. Thanks to their input we have made the following improvements to the paper during the rebuttal period:

- Corrected the title in the pdf so that it now corresponds to the submission title (Page 1\)
- Clarified the contributions of the paper by adding a list at end of introduction (end of page 2\)
- Clarified the motivation leading up to the list of harmonized synthetic data qualities that we use for our study (previously line 308 to line 325, now on lines 325 to 329 on page 7\)
- Extended analysis of the generalization result, comparing the generalization result to result in Table 1 where models have access to full data (page 8, under section 5.2), i.e. the baseline comparisons for Generalization experiment (5.2) and K-Anonymity experiment (5.3)
- Extended the evaluation by also using DiffTraj’s evaluation measures to Table 1 (Page 9), briefly mentioning them in the Evaluation section and given a more detailed description in A.3.3 (Appendix)
- Added an ablation study that isolates the improvement of using local information (Text on page 8-9 and data in Table 4 on page 10\)

Errata: Section 5.3.1 should be 5.4

---

### Meta-Review · Area_Chair_mk6M · 2024-12-19

**Metareview:**

This paper proposes a spatio-temporal generative model for trajectories, TDDPM, which focuses primarily on trajectories of peoples' movement in cities.  Although there are several interesting ideas, the reviewers also raised several concerns on evaluation, contribution, and privacy-preserving properties.  In particular, the paper is not thoroughly written because it is not aligned with the title about privacy.  Thus, I recommend a reject.

**Additional Comments On Reviewer Discussion:**

Reviewer 3LhK pointed out the misalignment with the previous working title.  Through multiple rounds of discussion, we reached the consensus that the paper needs a major revision.  The authors also acknowledged their mistake.

---

### Decision · Program_Chairs · 2025-01-22

Reject